# The OOD Blind Spot of Unsupervised Anomaly Detection

**Matthäus Heer**                                                    MATTHAEUS.HEER@UZH.CH
*Physik-Institut, University of Zurich, Winterthurerstrasse 190, 8057 Zurich, Switzerland*
**Janis Postels**                                                    JPOSTELS@VISION.EE.ETHZ.CH
**Xiaoran Chen**                                                     CHENX@VISION.EE.ETHZ.CH
**Ender Konukoglu**                                      ENDER.KONUKOGLU@VISION.EE.ETHZ.CH
*Computer Vision Lab, ETH Zurich, Sternwartstrasse 7, 8092 Zürich, Switzerland*
**Shadi Albarqouni**                              SHADI.ALBARQOUNI@HELMHOLTZ-MUENCHEN.DE
*Helmholtz AI, Helmholtz Zentrum München, Ingolstädter Landstraße 1, 85764 Neuherberg, Germany*
*AI in Medicine, Technical University of Munich, Einsteinstraße 25, 81675 Munich, Germany*

## Abstract

Deep unsupervised generative models are regarded as a promising alternative to supervised counterparts in the field of MRI-based lesion detection. They denote a principled approach for detecting unseen types of anomalies without relying on large amounts of expensive ground truth annotations. To this end, deep generative models are trained exclusively on data from healthy patients and detect lesions as Out-of-Distribution (OOD) data at test time (i.e. low likelihood). While this is a promising way of bypassing the need for costly annotations, this work demonstrates that it also renders this widely used unsupervised anomaly detection approach particularly vulnerable to non-lesion-based OOD data (e.g. data from different sensors). Since models are likely to be exposed to such OOD data in production, it is crucial to employ safety mechanisms to filter for such samples and run inference only on input for which the model is able to provide reliable results. We first show extensively that conventional, unsupervised anomaly detection mechanisms fail when being presented with true OOD data. Secondly, we apply prior knowledge to disentangle lesion-based OOD from their non-lesion-based counterparts.

## 1. Introduction

MRI (Magnetic Resonance Imaging) scans pose the primary screening method to detect, assess and segment brain pathologies for diagnosis and subsequent treatment planning. While supervised Deep Learning approaches contribute state-of-the-art lesion segmentation techniques [(Valverde et al., 2017), (Menze et al., 2014)], they are constrained to the distribution of anomalies used during training and the need for corresponding pixel-wise ground-truth labels from domain experts. This process is expensive and subject to an inter-/intra-rater ambiguity (Moraal et al., 2010). Unsupervised methods on the other hand have the potential to serve as a class-agnostic anomaly detection framework and might act as a quality assurance tool for practitioners without the need to curate large specialized training datasets.

Nevertheless, unsupervised anomaly detection techniques exploit OOD detection as their working principle, making them vulnerable to actual OOD samples during inference. Those are samples that might or might not contain lesions but more importantly originate from

---

\* S.A. was with Computer Vision Lab, ETH Zurich at the time of proposing this project.

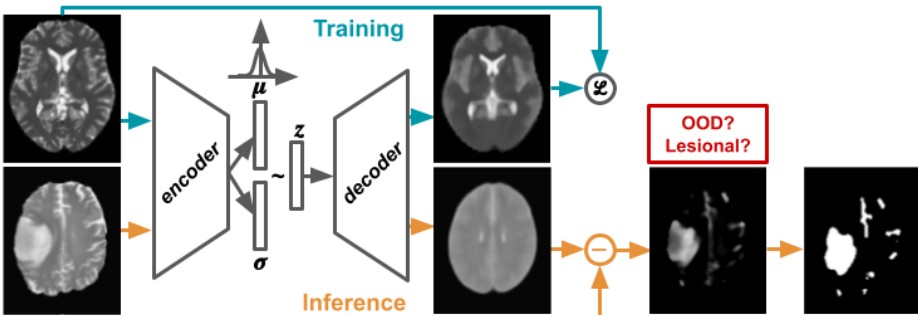

Figure 1: Working principle of unsupervised lesion detection based on Variational Autoencoders (VAEs) as used in this work. During training, a VAE learns to approximate the distribution of healthy images by maximizing the so-called Evidence Lower Bound (ELBO) $\mathcal{L}$, cf. Equ. 1. During inference, the residual map of an input test image yields the pixel-wise lesion detection map. Since this process inherently equates OOD detection with lesion detection, it is blind to domain-shift effects. Thus, disentangling the sources of abnormality of the image (OOD?, lesional?), be it due to a domain-shift or due to an actual lesion is of utmost importance and should be considered by default.

a different domain compared to the training distribution, which might be due to different scanner models used or changed parameters such as magnetic field strength. While the issue of unreliable model behavior for OOD samples is well-known and has received much attention in the case of supervised approaches in the past (Mårtensson et al., 2020), it is even more pronounced for unsupervised anomaly detection - which equates anomaly detection with OOD detection - by nature. Since models performing unsupervised anomaly detection in the field of MRI will be presented with data whose generating process is governed by numerous influencing factors that might provoke a domain-shift, this is of great concern for safe model deployment and is usually not addressed assuming that test samples follow the same distribution like the training data. Furthermore, datasets that contain both healthy and lesional samples are typically not publicly available which complicates the investigation of the severity of model performance degradation under such domain-shifts and makes this a particularly challenging problem since there is no access to lesional samples originating from the same domain. Thus, it is important to investigate whether common approaches can disentangle the underlying factors for a sample to be OOD - i.e. whether it is lesional or not. To this end, we also examine whether novel scores that arise from OOD detection using neural networks yield improvements with respect to finding the underlying cause.

**Unsupervised Lesion Detection**    Prior to the rise of Deep Learning-based approaches, works have been proposed to detect lesions in an unsupervised manner, such as registering images to a healthy standardized brain Atlas and fitting, amongst others, mixture models based on tissue-specific densities to detect lesions as model outliers (Kamber et al., 1995; Van, 2001; **?**). More recently, Deep Generative models based on Variational Autoencoder (VAE) (Kingma and Welling, 2014) and Generative Adversarial Networks (GAN) (Good-

fellow et al., 2014; Schlegl et al., 2017, 2019) have become popular due to their abilities to model high dimensional distributions, essentially learning what is referred to as the normative data distribution. During inference, anomaly detection is then performed by assessing the deviation of test samples from the training distribution. One common way within VAE-based frameworks is to do this via the reconstruction error of the reconstructed to the input sample. For images, pixel-level anomaly detection is performed by thresholding the residual map between an input and reconstruction images, as shown in Fig. 1, which builds upon the assumption that lesional regions are expected to have high reconstruction errors since they deviate from the training distribution. Recently, different metrics for pixel-level anomaly detection have been proposed by (Zimmerer et al., 2019). (Baur et al., 2018) introduced a VAE based framework incorporating adversarial training to improve the realism of reconstructions and avoid memorization. (Chen and Konukoglu, 2018) identified the lack of and improved latent space consistency by adding a regularizing constraint. (Chen et al., 2020) used a VAE with mixtures of Gaussian in the latent space, which is a more expressive prior distribution. At the same time, they applied Image Restoration on the reconstructed image prior to assessing the residual image. (Baur et al., 2021) presented a comprehensive comparative study for recent approaches to unsupervised lesion detection.

**Out-of-Distribution Detection** Unsupervised lesion detection techniques introduced above are based on the idea of detecting OOD samples using generative models since lesional samples do not fit the training distribution. There has been a rising interest in OOD detection, driven by the need to enable safe and interpretable model deployment since machine learning models usually perform inferior on OOD samples (Louizos and Welling, 2017; Goodfellow et al., 2014). Recently, (Mårtensson et al., 2020) raised awareness for this issue on medical MRI data in particular. Generative models seem to offer a principled approach to detecting OOD samples by applying a single-sided threshold on the data log-likelihood based on the training distribution (Bishop, 1994). However, recent work (Choi et al., 2019; Nalisnick et al., 2019a) has shown that generative models might assign a higher likelihood to OOD data than to in-distribution data, rendering this method problematic. The latter also proposed the so-called Watanabe–Akaike information criterion (WAIC) score which gives an asymptotically correct estimate of the gap between the training set and test set expectations. While this metric does not address the notion of typicality like (Nalisnick et al., 2019b), that is, assessing where the largest amount of probability mass resides within a high-dimensional feature space, it works surprisingly well in practice. Recently, (Morningstar et al., 2020) introduced another density-based OOD detection framework by aggregating various inference statistics, e.g. the reconstruction errors, into the so-called Density of States Estimation (DoSE) score. Specifically, they fit a Kernel Density Estimator (KDE) to each statistics distribution evaluated on the training data and mark novel samples as OOD by thresholding their sum of likelihoods under said estimators.

**Contributions** This work raises awareness of the issue that the predominant approach to unsupervised lesion detection is particularly vulnerable to OOD samples. This is done by assessing multiple common metrics for anomaly and OOD detection and concluding that predictions don't reflect the true underlying reason for a sample to be labeled abnormal. While this work does not aim to provide state-of-the-art lesion segmentation performance, we explore concepts originating from recent work in OOD detection for Deep Generative

Models, to enhance model robustness when presented with OOD data. More precisely, we deploy and adapt recent approaches for OOD detection to answer the following questions: Are OOD detection metrics suitable for sample-wise anomaly detection and is it possible to disentangle lesion-based OOD samples from their non-lesion-based counterparts? Finally, we explore the use of prior knowledge in the form of the entropy on the residual map in an attempt to disentangle the influencing factors of lesions and domain-shifts during inference.

## 2. The OOD Blind Spot of Unsupervised Anomaly Detection

The following describes the theoretical foundations of the unsupervised lesion detection framework based on VAE and its limitations due to entanglement of domain-shift and lesion effects before finally discussing proposals on how to overcome those difficulties. Note that while we discuss the limitations here on the of VAE-based approaches, these also extrapolate to other unsupervised methods (Schlegl et al., 2019) that equate anomaly detection with OOD detection. The objective of unsupervised lesion detection (*cf.* Fig. 1) is to train a generative model $\mathbf{f}_\theta(\cdot)$ on a set of healthy images $\mathbf{X} = \{\mathbf{x}^{(i)}\}_{i=1}^N$, where $\mathbf{x}^{(i)} \in \mathbb{R}^{m \times n}$, to predict whether a query sample $\mathbf{x}_q^{(i)} \in \mathbb{R}^{m \times n}$ is anomalous, i.e. contains lesions (Sec. **??**), and to obtain a pixel-wise lesion map $\mathbf{l} \in \{0,1\}^{m \times n}$. We hypothesize that a sample might be predicted to be anomalous either due to actual lesions or due to a domain-shift which might cause the model to generate unreliable predictions since the commonly used metrics for anomaly detection do not differentiate those two sources.

### 2.1. Background

Using generative models to perform unsupervised anomaly detection has been widely adopted in tackling unsupervised lesion detection using MR images (Baur et al., 2021). Hereby, the predominant approach is to approximate the generally intractable data distribution $p(\mathbf{x})$ using the VAE framework for a set of healthy images $\mathbf{X}$.

VAEs (Kingma and Welling, 2014) aims to solve the generally intractable integral $p(\mathbf{x}) = \int p(\mathbf{x}|\mathbf{z})p(\mathbf{z})d\mathbf{z}$. It does so by introducing a surrogate posterior distribution $q(\mathbf{z}|\mathbf{x})$ and approximating the data log-likelihood $\log p(\mathbf{x})$ by maximizing the so-called *Evidence Lower Bound (ELBO)* $\mathcal{L}$

$$\log p(\mathbf{x}) \geq \mathcal{L} = \underbrace{E_{q_\phi(\mathbf{z}|\mathbf{x})}[\log p_\theta(\mathbf{x}|\mathbf{z})]}_{\text{Reconstruction Error}} - \underbrace{D_{KL}[q_\phi(\mathbf{z}|\mathbf{x})||p(\mathbf{z})]}_{\text{Prior Loss}}, \tag{1}$$

where $q_\phi(\mathbf{z}|\mathbf{x})$ and $p_\theta(\mathbf{x}|\mathbf{z})$ are modelled as neural networks with parameters $\phi$ and $\theta$, respectively. The former encodes input samples into the latent space $\mathbf{z}$ and thus is denoted as the *encoder*. The latter is trained to reconstruct the input from this latent representation and hence is called the *decoder*. This framework allows for posterior inference via the learned approximate posterior distribution $q_\phi(\mathbf{z}|\mathbf{x})$ which is being pushed towards a prior $p(\mathbf{z})$, $\mathcal{N}(\mathbf{0}, I)$ in our case, during training by minimizing their KL Divergence denoted as the prior loss.

Using a VAE trained on $p(\mathbf{x})$, a potentially lesional test sample $\tilde{\mathbf{x}}^{(i)}$ is being fed through the generative model to obtain a reconstruction $\hat{\mathbf{x}}^{(i)}$. Being trained on healthy samples exclusively, the model is expected to not be able to reconstruct lesional components while it

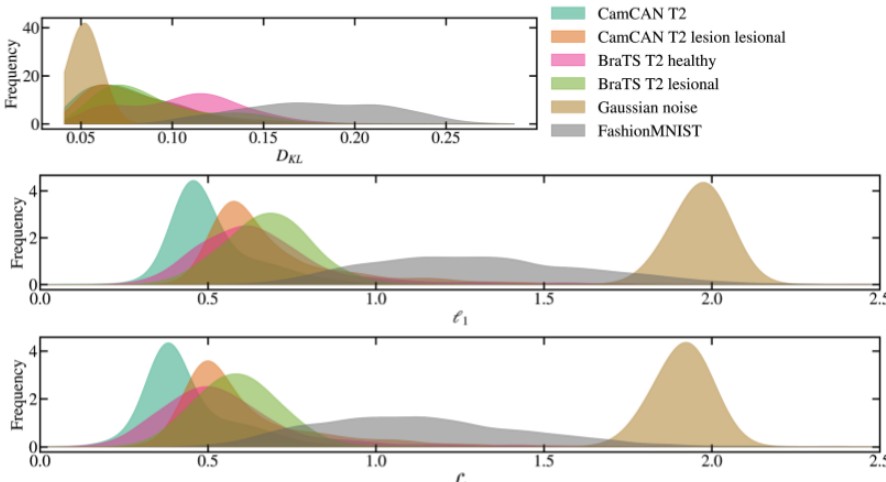

Figure 2: Densities for sample-wise loss term contributions (cf. Equ.1) for various OOD datasets. *CamCAN T2* (teal) represents the in-distribution training data containing only healthy slices. *CamCAN T2 lesion* holds samples from *CamCAN T2* but with artificially added Gaussian blobs to simulate lesional samples as explained in Sec. 3 and shown in Fig. 3. All other datasets can be regarded as OOD. We can conclude that non of the commonly used metrics, that is, $D_{KL}$, $l_1$ (reconstruction error) or $\mathcal{L}$ is able to differentiate between whether a sample is being detected as abnormal due to a domain-shift or due to actual lesions.

should succeed in reconstructing healthy parts of the input. A pixel-wise anomaly segmentation map can thus be retrieved by thresholding the pixel-wise residual $\mathbf{r} = \|\hat{\mathbf{x}}^{(i)} - \tilde{\mathbf{x}}^{(i)}\|_p \in \mathbb{R}^{m \times n}$, where $\|\cdot\|_p$ is the $\ell_p$-norm, and $p$ is chosen to be 1. A pixel is being marked as anomaly if the per-pixel value in the residual image $\mathbf{r} = \|\hat{\mathbf{x}}^{(i)} - \tilde{\mathbf{x}}^{(i)}\|_p$ exceeds some threshold $\tau$ (*cf.* Sec. 5).

### 2.2. OOD Detection

For a given dataset $\{\mathbf{x}^{(i)}\}_{i=1}^N, \mathbf{x}^{(i)} \in \mathbb{R}^{m \times n}$, sampled from a distribution $p_{data}(\mathbf{x})$, OOD detection aims to answer the question whether a novel sample $\mathbf{x}^{(i)}$ is sampled from the same data generating distribution $p_{data}(\mathbf{x})$ or some other unknown distribution.

At test time, the model might be exposed to samples from the following three categories cf. Fig. 2:

 (i) **healthy & in-distribution**, anomaly-free images from the same domain as the training data (e.g. CamCAN T2)
 (ii) **healthy & OOD**, anomaly free-images with domain shift from the training data (e.g. BraTS T2 healthy),
(iii) **lesional & OOD**, images with lesions regardless of domain shift (e.g. BraTS T2 lesional),
.

To gain an intuition for the capabilities of each OOD score, we first assess their overall capability to distinguish between in- and out-of-distribution samples. Interestingly, we find that one recently proposed score indeed outperforms all commonly used metrics in unsupervised lesion detection. The following metrics act as OOD scores for which we report the area under ROC $AU_{ROC}$ and PRC $AU_{PRC}$ curves. (1) the mean reconstruction error $\ell_1$ per pixel for a whole sample, (2) the KL divergence $D_{KL}$ between posterior and prior and (3) the ELBO $\mathcal{L}$ from Equ. 1. Furthermore, we exploit recent OOD metrics, namely the (4) $WAIC = E_\theta[p_\theta(\mathbf{x})] - Var_\theta[\log p_\theta(\mathbf{x})]$ score (Choi et al., 2019), where $\theta$ denote model parameters of an ensemble of models, and (5) the $DoSE = \sum_j KDE_j(\mathbf{x})$ (Morningstar et al., 2020) score. For $DoSE$, metrics (1)-(3) are being used as training statistics.

## 2.3. Disentangling Lesional and Non-Lesional OOD Samples

Our results (cf. Sec. 3.1) suggest that these classical OOD detection scores are incapable of discriminating between healthy and lesional samples which is in agreement with the findings from Fig. 2. However, the $DoSE$ score offers the possibility to craft statistics which potentially help to disentangle OOD scores for samples originating from groups (i) - (iii).

**Assumption** Lesional samples are expected to have large residual errors confined in relatively small regions considering pixels within a tumor dominating the reconstruction error. On the other hand, a (healthy) OOD sample should show a steady but spread out error due to global domain-shift effects which is equivalent to a large uncertainty in the pixel-wise error distribution which might be captured by the following entropy measure. We extend the framework with an entropy statistic $H_{\ell_1}$ and investigate its suitability to disentangle the underlying reasons for which a sample appears to be marked OOD. The normalized sample-wise entropy scores are calculated on a normalized residual map $\mathbf{r}$ via $H_{\ell_1}(\mathbf{r}) = -\sum_j^{n_p} \frac{\mathbf{r}_j \log \mathbf{r}_j}{\log n_p}$. where $n_p$ is the number of pixels within the brain mask.

To conclude, our framework provides two scores, a global OOD score in the form of $WAIC$ or $DoSE$ and a second score using $H_{\ell_1}$ to address the problem of slice-wise anomaly detection in particular.

## 3. Experiments

**Datasets.** Brain scan MRI training samples from 653 healthy patients originate from the CamCAN dataset (Taylor et al., 2017) and only T2 weighted MR images are used for training. For detection, we use T2- (and T1-weighted) images from the BraTS 2017 dataset (Menze et al., 2014) which contains high- and low-grade glioma samples and provides pixel-level ground-truth lesion segmentations. BraTS is considered to be OOD w.r.t. domain-shift due to different scanning parameters and models. Since, to the best of our knowledge, there exist no public MR brain datasets holding both healthy and lesional samples, we artificially crafted lesional samples on top of the in-distribution validation set as described in Sec. 5.3. We also test our algorithms on OOD datasets of non-medical images, which in theory reside far away from the data distribution of group (i), e.g., MNIST and images of Gaussian noise. Finally, to assess whether the widely used practice of histogram matching mitigates this problem, we created matched and un-matched versions of the BraTS datasets. For

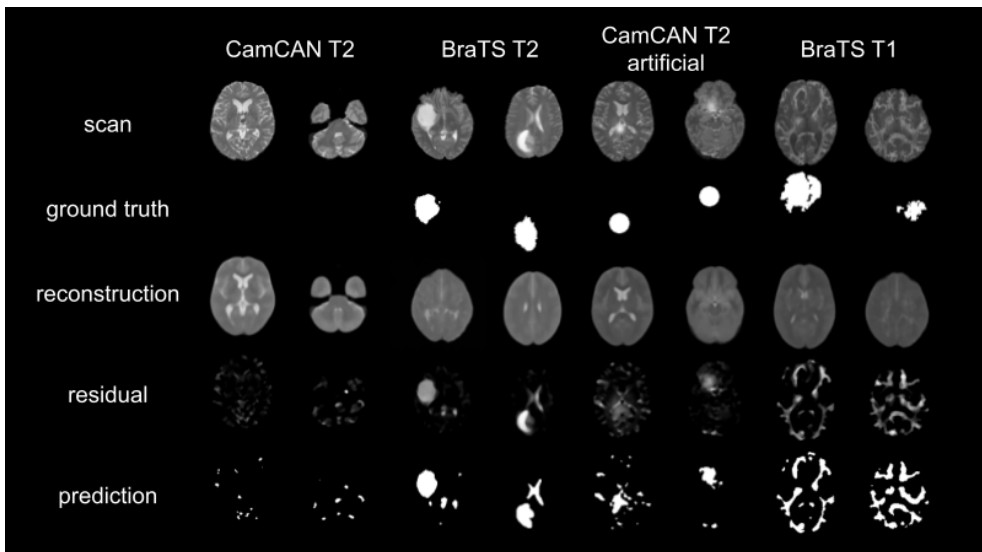

Figure 3: Examples of pixel-wise anomaly detection pipeline using various OOD datasets. The model has been trained on *CamCAN T2*. Columns 1-2 show For *CamCAN T2 artifical*, the lesions have been generated artificially using 2D Gaussian blobs where the ground truth segmentation refers to the $1\sigma$ intensity drop.

all tables, *HM* stands for matched histograms, where the histogram of each subject in CamCAN and BRATS is matched to that of a reference subject randomly selected from CamCAN. As histogram matching can be seen as a naive way to reduce the domain shift, we investigate whether it leads to an accuracy gain when detecting anomalous slices from different domains.

### 3.1. OOD Detection and Sample-wise Anomaly Detection

Tab. 1 reveals that the $WAIC$ score performs best at distinguishing between in-distribution and OOD slices regardless of whether the OOD samples contain lesions. It outperformed the $DoSE$ approach (using $l_1$, $D_{KL}$ and $\mathcal{L}$ statistics) which is seemingly affected by the $D_{KL}$ term suffering from performance degradation. Thus, the $WAIC$ score may serve as a strong candidate to perform initial filtering for OOD samples before employing a lesion-agnostic score to distinguish healthy from lesional slices. Tab. 2 on the other hand shows results for classifying slices as healthy or lesional. The positive class is chosen to hold lesional slices from the test set while the negative class consists of healthy slices from the in-distribution training data as well as healthy slices from the test set. While we have found the $D_{KL}$-term to be a strong candidate metric for anomaly detection in some settings (similar to (Zimmerer et al., 2019)), we have found it to be heavily influenced by architectural choices and training dynamics which let's us conclude that it is not a robust choice for metric for the task at hand. The proposed normalized entropy score shows no more discriminative power compared to the established metrics (such as $l_1$) which demands more investigation into ways to disentangle the underlying mechanisms for a sample to be OOD or not.

Table 1: OOD detection performance. The negative class holds samples from CamCAN T2 while the positive class consists of samples from the respective OOD dataset. OOD samples are subdivided s.t. either only healthy, lesional or all of them are considered. $\ell_1$ is the mean reconstruction error per pixel for a slice, $D_{KL}$ the KL divergence between the prior and approximate posterior and $\mathcal{L}$ the ELBO. We further introduce recent metrics from OOD detection in the form of $WAIC$ and $DoSE$ score using $\ell_1$, $D_{KL}$ and $\mathcal{L}$ training statistics. We observe that $WAIC$ performs best at detecting domain-shifted samples across all datasets.

| OOD Metric | | $\ell_1$ | | $D_{KL}$ | | $\mathcal{L}$ | | $WAIC$ | | $DoSE_{(l_1,D_{KL},\mathcal{L})}$ | |
|---|---|---|---|---|---|---|---|---|---|---|---|
| OOD Dataset | | AU$_{ROC}$ | AU$_{PRC}$ | AU$_{ROC}$ | AU$_{PRC}$ | AU$_{ROC}$ | AU$_{PRC}$ | AU$_{ROC}$ | AU$_{PRC}$ | AU$_{ROC}$ | AU$_{PRC}$ |
| BraTS | all | 0.89 | 0.86 | 0.68 | 0.61 | 0.85 | 0.81 | 0.93 | **0.95** | 0.78 | 0.70 |
| T2 | healthy | 0.85 | 0.84 | 0.73 | 0.71 | 0.79 | 0.79 | 0.87 | **0.91** | 0.78 | 0.76 |
| HM | lesion | 0.92 | 0.89 | 0.63 | 0.55 | 0.90 | 0.87 | 0.99 | **0.99** | 0.78 | 0.70 |
| BraTS | all | 0.87 | 0.85 | 0.69 | 0.62 | 0.83 | 0.81 | 0.91 | **0.92** | 0.79 | 0.72 |
| T2 | healthy | 0.84 | 0.83 | 0.74 | 0.72 | 0.77 | 0.78 | 0.84 | **0.85** | 0.80 | 0.77 |
| | lesional | 0.92 | 0.90 | 0.64 | 0.56 | 0.89 | 0.86 | 0.98 | **0.98** | 0.78 | 0.70 |
| BraTS | all | 0.92 | 0.91 | 0.73 | 0.64 | 0.88 | 0.88 | 0.92 | **0.95** | 0.82 | 0.76 |
| T1 | healthy | 0.88 | 0.88 | 0.76 | 0.71 | 0.83 | 0.85 | 0.86 | **0.91** | 0.82 | 0.79 |
| HM | lesional | 0.96 | 0.95 | 0.70 | 0.60 | 0.95 | 0.93 | 0.99 | **0.99** | 0.83 | 0.76 |
| BraTS | all | 0.95 | 0.94 | 0.77 | 0.74 | 0.91 | 0.90 | 0.97 | **0.98** | 0.86 | 0.82 |
| T1 | healthy | 0.94 | 0.94 | 0.82 | 0.82 | 0.88 | 0.88 | 0.94 | **0.96** | 0.88 | 0.86 |
| | lesional | 0.97 | 0.96 | 0.73 | 0.66 | 0.96 | 0.96 | 1.00 | **1.00** | 0.84 | 0.78 |
| CamCAN | all | 0.69 | 0.69 | 0.55 | 0.58 | 0.66 | 0.68 | 0.74 | **0.79** | 0.63 | 0.61 |
| T2 | healthy | 0.51 | 0.51 | 0.51 | 0.50 | 0.52 | 0.52 | 0.68 | **0.73** | 0.52 | 0.50 |
| artificial | lesional | 0.87 | **0.85** | 0.60 | 0.64 | 0.84 | 0.82 | 0.81 | 0.84 | 0.74 | 0.70 |
| MNIST | | 1.00 | 1.00 | 0.00 | 0.31 | 1.00 | 1.00 | 1.00 | 1.00 | 1.00 | 1.00 |
| Gaussian Noise | | 1.00 | 1.00 | 0.00 | 0.30 | 1.00 | 1.00 | 1.00 | 1.00 | 1.00 | 1.00 |

Table 2: Slice-wise anomaly detection on various Out-of-Distribution datasets. The positive class holds lesional slices from the respective test set while the negative class holds healthy slices from the test and in-distribution validation set (CamCAN T2). To this point, neither the entropy score nor classical OOD detection metrics are able to outperform the classical reconstruction error metric.

| Metric | $\ell_1$ | | $D_{KL}$ | | $\mathcal{L}$ | | $H_{\ell_1}$ | | WAIC | | DoSE($\ell_1, D_{KL}, \mathcal{L}, H_{\ell_1}$) | |
|---|---|---|---|---|---|---|---|---|---|---|---|---|
| Test Set | AU$_{ROC}$ | AU$_{PRC}$ | AU$_{ROC}$ | AU$_{PRC}$ | AU$_{ROC}$ | AU$_{PRC}$ | AU$_{ROC}$ | AU$_{PRC}$ | AU$_{ROC}$ | AU$_{PRC}$ | AU$_{ROC}$ | AU$_{PRC}$ |
| BraTS T2 HM | 0.74 | **0.44** | 0.50 | 0.31 | 0.74 | **0.44** | 0.66 | 0.37 | 0.64 | 0.41 | 0.55 | 0.30 |
| BraTS T2 | 0.72 | 0.41 | 0.47 | 0.29 | 0.73 | **0.42** | 0.65 | 0.35 | 0.62 | 0.39 | 0.53 | 0.29 |
| BraTS T2 HM H-flip | 0.73 | **0.41** | 0.50 | 0.30 | 0.73 | **0.41** | 0.66 | 0.36 | 0.62 | 0.38 | 0.54 | 0.29 |
| BraTS T2 HM V-flip | 0.59 | 0.31 | 0.38 | 0.22 | 0.59 | 0.31 | 0.53 | 0.27 | 0.62 | **0.36** | 0.46 | 0.25 |
| CamCAN T2 artificial | 0.83 | **0.82** | 0.53 | 0.61 | 0.82 | **0.82** | 0.71 | 0.75 | 0.77 | 0.78 | 0.67 | 0.66 |

## 4. Conclusion

In this work, we have shed light on the issue that unsupervised lesion detection and Out-of-Distribution detection are by nature entangled since, by definition, the former relies on the latter. We point out that more work needs to be done to fully understand and solve the issue of entangled OOD factors to enable safe deployment of models performing unsupervised lesion detection, such as further analysis with other types of detection frameworks, i.e. restoration-based methods.

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

## 5. Appendix

### 5.1. Implementation Details

We follow the generalised encoder-decoder architecture from (Baur et al., 2021). Each layer consists of a 2D convolution followed by batch normalisation and a LeakyReLU activation function. Models are trained for 80 epochs with a batch size of 128 and linear $\beta$-annealing (weight of KL-term) from 0.0 to 0.3 over the first 5 epochs. The final value for $\beta$ has been found empirically by maximizing reconstruction performance (low $\beta$ better) while still producing visually coherent samples. Adam optimizer is used with an initial learning rate of $10^{-4}$. The full source code will be made publicly available at https://github.com/matthaeusheer/uncertify.

### 5.2. Pixel-wise Anomaly Detection

**Determining the Pixel-wise Lesion Detection Threshold**  Since, in an unsupervised setting, there is no access to ground truth labels, it is not possible to tune such hyper-parameters with regards to the final metrics of interest, e.g. the Dice score. Instead, we follow the approach in (Konukoglu et al., 2018), where we assume any pixel from the training set images marked to be anomalous by the anomaly detection algorithm to be a false positive. We set a limit $l_{FPR}$ on the false positive rate $FPR_{train}$ we are willing to accept to determine a threshold that satisfies the constraint. The threshold is computed via the Golden Section Search algorithm (Kiefer, 1953) by solving the optimization problem $\tau = \min_t |FPR_{train}(t) - l_{FPR}|$. Finally, the threshold $\tau$ gets deployed to convert the residual map $\mathbf{r}$ to a binary lesion map for which the final segmentation metrics are being computed per patient on unseen test data.

**Pixel-Wise Anomaly Detection Performance**  Tab. 3 shows the pixel-wise anomaly detection performance of the model used throughout this study with and without post-processing steps in comparison with the baseline from (Chen et al., 2020). Ground truth is obtained from the lesion segmentation mask and only pixels within the brain mask are considered for this analysis. For clinical applicability, Dice scores are calculated per patient reporting mean and standard deviation. It is evident that performing histogram matching prior to inference improves lesion detection performance slightly. Comparison is being made with the state-of-the-art performance (Chen et al., 2020) since they use the same training and test datasets. Our results cf. Tab. 3 outperform the baseline mainly due to post-processing applied but lack behind the state-of-the-art which implements a more powerful latent space representation and image restoration.

### 5.3. Preprocessing and Postprocessing steps

**Preprocessing**  Brain scan MRI training samples from healthy patients originate from the CamCAN dataset (Taylor et al., 2017) and only T2 weighted MR images are used for training. Pre-processing includes bias correction using the N4 algorithm (Tustison et al., 2010) and centering of the brain. Patient-wise histogram matching to a randomly chosen in-distribution sample ensures similar intensity profiles throughout all training samples. Finally, all pixel values within the brain mask are normalized to zero mean and unit variance,

Table 3: Pixel-wise anomaly segmentation (Dice) and detection ($AU_{ROC}$ / $AU_{PRC}$) performance. [*] includes post-processing (smoothing & mask-erosion of residual map) , [**] no post-processing, [***] results from (Chen et al., 2020).

| Dataset | BraTS T2 HM | | | BraTS T2 | | | CamCAN T2 artificial lesions | | | BraTS T1 | | |
|---|---|---|---|---|---|---|---|---|---|---|---|---|
| Model | Dice | $AU_{ROC}$ | $AU_{PRC}$ | Dice | $AU_{ROC}$ | $AU_{PRC}$ | Dice | $AU_{ROC}$ | $AU_{PRC}$ | Dice | $AU_{ROC}$ | $AU_{PRC}$ |
| Baseline[***] | $0.23 \pm 0.13$ | 0.69 | - | - | - | - | - | - | - | - | - | - |
| Ours VAE[*] | $0.34 \pm 0.12$ | 0.75 | 0.25 | 0.31 | 0.73 | 0.20 | 0.63 | 0.92 | 0.66 | 0.10 | 0.51 | 0.07 |
| Ours VAE[**] | $0.25 \pm 0.11$ | 0.69 | 0.16 | 0.22 | 0.66 | 0.13 | 0.39 | 0.88 | 0.43 | 0.10 | 0.49 | 0.07 |
| GMVAE[***] | $0.46 \pm 0.23$ | 0.83 | - | - | - | - | - | - | - | - | - | - |

again, on a per-patient level. Empty MR slices are excluded during training. To obtain lesional samples following the same distribution as CamCAN T2, we artificially crafted lesional samples by randomly adding high-intensity Gaussian blobs to CamCAN T2-weighted images. The blobs standard deviations range from 0 to 10 (given images of size $128 \times 128$) and are weighted such as to have similar maximum intensities compared to real lesions. Finally, we applied vertical and horizontal flipping (V-flip, H-flip) following common practices in recent OOD detection works.

**Post-Processing** Postprocessing, that is eroding the brain mask inwards and applying median filtering on the residual map to reduce false positives, yields significant performance boosts. We did not perform 3D connected component filtering which is expected to increase performance even further (Baur et al., 2021).

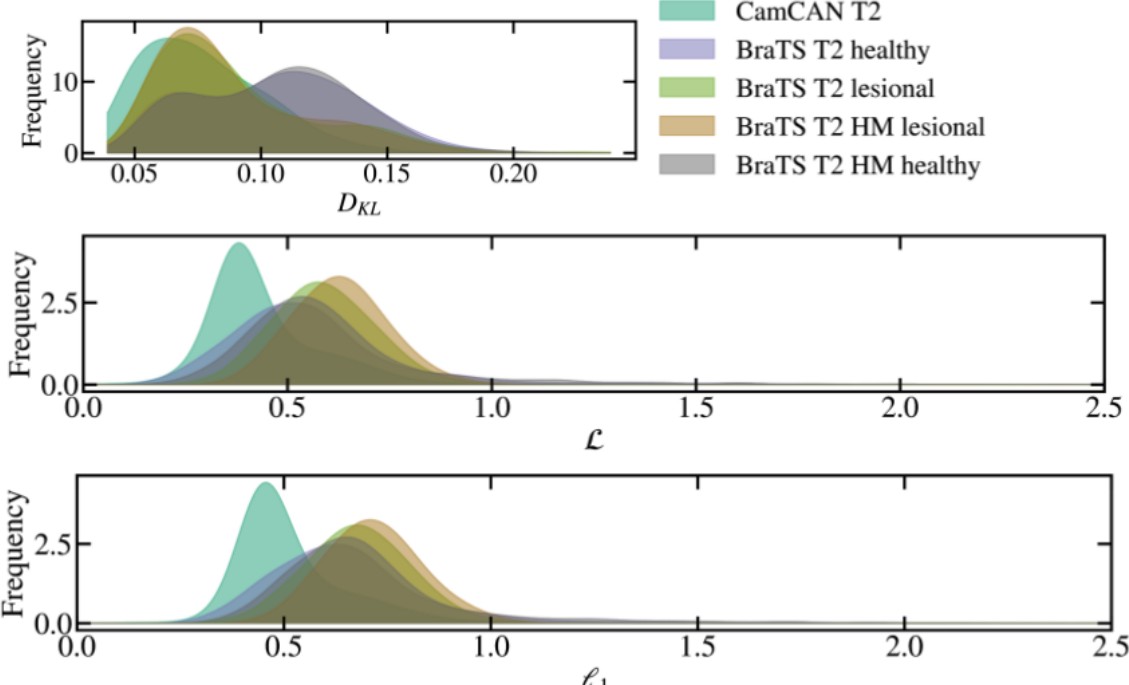

Figure 4: Similar to Fig. 2, however, now the impact of *histogram matching* before inference shall be emphasized. While histogram matching generally improves the reconstruction and lesion segmentation performance (cf. Tab. 3), it does not provide a solution to the entanglement of domain-shift and lesions. That is, it does not shift the healthy OOD samples in a way that their distributions overlap with the healthy in-distribution samples.

## 5.4. Volumetric Consistency of the Reconstruction Error

One might be tempted to question whether it is enough to consider individual slices in the anomaly detection process without leveraging the information of slices nearby. That is, could it happen that a slice predicted to be healthy might be embedded in two slices which are predicted to be lesional. Fig. 5 reveals that this is typically an unrealistic scenario for the case of hyper-intense lesions. Nevertheless, 3D convolutional neural networks might be an interesting way forward, especially when the lesions are not so distinct as the one shown in the example.

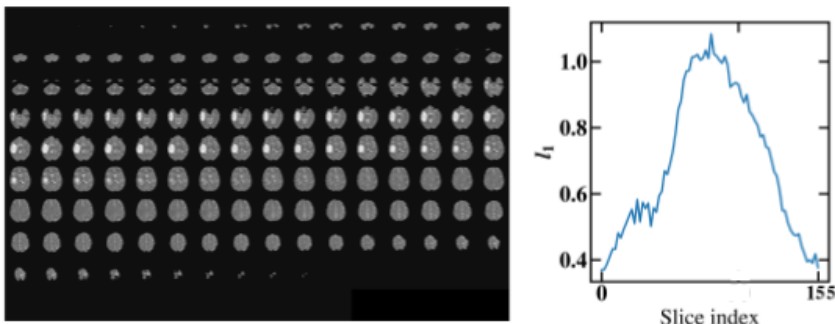

Figure 5: Left: All axial slices comprising a single BraTS T2 sample. Right: Corresponding slice-wise mean per-pixel reconstruction error, $l_1$. This reveals that the reconstruction error is close to being smooth and slices marked as healthy embedded in lesional slices can be expected to be rather unlikely.

