# OpenReview forum: "The OOD Blind Spot of Unsupervised Anomaly Detection"
_MIDL.io/2021/Conference — MIDL 2021_

### Official Review · AnonReviewer4 · 2021-03-03

**Confidence:** 4
**Preliminary Rating:** 2
**Recommendation:** Poster
**Final Rating:** 3

**Summary:**

The paper at hand addresses the problem of OOD-based lesion detection where a deep learning model learns a representation of healthy data and lesions are segmented by using an erroneous reconstruction of diseased cases. The authors focus on the problem of distinguishing between two types of OOD samples: 1) OOD characteristics due to the scans having lesions 2) OOD characteristics due to domain shifts (different scanners/protocols). The authors highlight that this problem is entangled and that current OOD-based lesion detectors cannot distinguish between these two types of OOD cases. This is highlighted with experiments on public datasets. The authors try to improve the differentiation of the two OOD cases with an entropy measure. However, this measure does not appear to be helpful.

**Strengths:**

•	The paper is well-written

•	The authors provide an extensive literature overview

•	The authors identify an interesting problem in the field of OOD detection

•	The authors present results that highlight the problem of distinguishing between actual lesions and domain OOD samples for OOD-based lesion detection

•	The authors provide results on several public datasets


**Weaknesses:**

•	The authors highlight an interesting problem, however, neither a working solution nor any method showing improvement is presented. The authors make an attempt at using an entropy score to find distinguish healthy and lesional OOD samples but the method does not appear to work. One could argue that the contribution of highlighting the problem alone is limited.

•	Although the authors focus on highlighting the problem of distinguishing between healthy and lesional OOD detection, they do not discuss the implications for real-world application. What is the actual danger? Wouldn’t an OOD-based lesion detection method simply have a larger false-positive rate, as it flags more scans as OOD than necessary? Is this really a special problem or just the “usual” OOD problem?
Following up on that – wouldn’t it be a more severe problem if these methods detected OOD lesional images as in-distribution healthy? However, looking at the results in Table 1 (OOD lesional), the methods do not appear to have a problem with this scenario. Leading once again to the question – are domain-related OOD samples really a big problem for OOD-based lesion detectors?

•	A big part of the authors’ results is in the appendix. Therefore, the amount of content in the paper itself is limited, partly, due to lengthy descriptions of basics and background. I would recommend shortening the background (2/2.1) and including more results in the paper itself.


**Deanonymize Review:**

no

**Detailed Comments:**

•	Section 2: “Note that while discuss the limitations” is missing a word

•	Section 3.1: The authors state “the negative class consists of healthy slices from the in-distribution training data as well as healthy slices from the test set.” In their text and they state “the negative class holds healthy slices from the test and in-distribution validation set” in the caption of Table 2. The authors should clearly state what they use – I assume it is the in-distribution validation data

•	In Table 1, the author list CamCAN healthy as an OOD dataset. How should this be understood? Based on the authors’ description I assumed that this is the in-distribution dataset. This should be explained more clearly.


**Final Rating Justification:**

The authors addressed the two points I raised, as requested. The first limitation I mentioned in terms of weaknesses remains (and cannot really be fixed), therefore my final rating is weak accept.

**Justification Of The Preliminary Rating:**

See weaknesses. While I agree that the problem is interesting, I do not see an immediate problem for the clinical application of OOD-based methods. I am open to being convinced otherwise. Besides, the authors only highlight the problem but provide no solution or any improvement to the problem that was highlighted.

**Paper Type:**

validation/application paper

**Questions To Address In The Rebuttal:**

I agree that the problem is interesting from a scientific perspective. However, it is not quite clear to me, how this is a practical problem for the clinical application of these methods. I recommend the authors provide arguments on this point.

Furthermore, I would recommend shortening the background and adding results from the appendix to the actual paper. Otherwise, the content would be rather limited.


**Special Issue:**

no

---

### Official Review · AnonReviewer3 · 2021-03-08

**Confidence:** 5
**Preliminary Rating:** 3
**Recommendation:** Oral

**Summary:**

The authors proposed a detailed study for the entanglement for the lesion and non-lesion out of distribution data distribution. This issue happens due to the domain shift between different scan sources ( different scanners ,.... ). They provided detailed experiments to backup their work. They also highlighted on the effect of the histogram matching as similar approach to domain adaptation.

**Strengths:**

They provided a detailed study to the out of distribution unsupervised method for lesion detection. They studied different loss terms and to show the effect on the quantitative metrics. Also, they studied the effect of domain shift on the anomaly detection process.

**Weaknesses:**

- The authors simulated the lesion using high-intensity gaussian noise. The validation process of this assumption assumption is not discussed other than "artificially crafted lesional samples ".  Details about the assumption can clarify if the gaussian noise matches the original data distribution or it will just considered as anomaly all the times.

**Deanonymize Review:**

no

**Detailed Comments:**

- The colours in Fig.2 can overlap and the real distribution can be challenging to follow if the are only lines not filled may be easier to see.
- The effect of the histogram shall be showed in Fig. 2 sub-plot as it is one of the main points that has to be emphasised.
- Fig. 3 can be beneficial if moved to the text and included in the discussion.
- The effect on Fashion MNIST is not obvious specially it has different data distribution from the application.

**Justification Of The Preliminary Rating:**

The proposed work addressed the effect of domain shift on the anomaly detection. The paper discussed the disentanglement process in the detection process for out-of-detection distribution. They based their work on two datasets that have domain shift effect between them.

**Paper Type:**

validation/application paper

**Special Issue:**

no

---

### Official Review · AnonReviewer1 · 2021-03-08

**Confidence:** 4
**Preliminary Rating:** 4
**Recommendation:** Oral
**Final Rating:** 4

**Summary:**

This paper looks into anomaly detection by means of OOD analysis in MRI brain scans, attempting to understand if current standard approaches truly work as they promise: can they really understand if a sample is OOD because it contains a lesion or not? To ascertain this question, the authors add a second potential cause for being considered OOD, which is coming from a different domain but not containing a lesion, and explore if current techniques mark that as an anomaly. They do they analysis at the scan, slice, and pixel level, also adapting recent OOD scores to this task in order to try to disentagle the lesion factor from the out-of-domain factor, which seems to improve results.

**Strengths:**

The paper is very well written, and I find it very honest. It is important to reflect on the actual purposes of anomaly detection techniques in this context, and test them on realistic conditions rather than on super-controlled data, and the authors make a nice attempt in that direction. The contextualization of the work within the broader area is nicely explained, and the experiments seem well posed. I see no need for state-of-the-art results in this kind of articles, I find it very interesting as it is, congratulations.

**Weaknesses:**

- I have what may be a naïve question. If I have a source dataset of healthy images, I train an anomaly detection model, hoping to be able to reject scans that contain lesions, and then I feed my model with scans coming from a different domain, what am I expecting to happen? I mean, wouldn't the ideal behavior be that the system works the same, and flags scans with lesions, and does not flag scans without them, regardless of the domain? Because, if I am right, the message in the paper seems to be that those scans should be flagged as OOD regardless of containing a lesion or not. Then the authors go ahead and show that current anomaly detection systems are not consistent when faced with data from other domains, which is fine, but isn't the whole story a bit incomplete? Could the authors clarify this, please?

- Another question would be, I understand that all this is trained on 2d data, which kind of loses the information of two slices being consecutive, so to speak. Shouldn't that be leveraged in some way to increase OOD performance? Like, I would not expect to see two OOD slices with an in-distribution slice in the middle. Isn't this something to consider, even if it was in some sort of "not-so-elegant" post-processing step?

**Deanonymize Review:**

no

**Detailed Comments:**

It is probably beyond the scope of this work, and maybe it is not even possible at this point, but have the authors considered carrying part of their analysis on the recently held MICCAI20's MOOD challenge on OOD detection http://medicalood.dkfz.de/web/ ?

Aside from that, I only have minor/stylistic comments here:

- Maybe I did not get it, but what is the difference in Tables 1 and 2 between being boldfaced and being underlined? Also, the font size on tables is very unfriendly, I understand space constraints, but then maybe you should think of a different wait of laying out the results?

- Please check your references, there are things like this:
```
Ender Konukoglu and Ben Glocker. NeuroImage, 181:521–538, 2018. ISSN 10959572. doi:10.1016/j.neuroimage.2018.07.032.
```
Also, within the text there are a couple of citations to (Sch, 2019) which do not have a match in the references list, I think.

- In figure 2, I don't follow the "cf. 1" bit. Where should the reader head in order to understand which of the three sub-plots corresponds to what loss?

- " enable save and interpretable model deployment" -> safe?

- "while discuss the limitations here on the of VAE-based approaches" -> we discuss?

- "normalized (sums up to 1)" -> I guess everybody knows and you can skip the "sumps up to 1" bit?

- "by performing the optimization problem" -> by solving?

- "While BraTS is considered to be OOD w.r.t. domain-shift due to different scanning parameters and models." -> this sentence lacks a second part?

- " introducing an surrogate posterior distribution" -> a surrogate

-  "Post-processing (?), that is" -> is the interrogation mark a mistake?

**Final Rating Justification:**

I was already recommending strong accept for this work, and I am keeping that recommendation based not only on the answers provided to me by the authors, but also on the replies to other reviewers.

**Justification Of The Preliminary Rating:**

A pleasure to read this paper, it is always important to challenge common beliefs on current approaches to ambiguously-posed problems like anomaly detection, and rigurously test them in alternative scenarios. I believe this work has a clear place in MIDL.

**Paper Type:**

methodological development

**Questions To Address In The Rebuttal:**

I am supporting acceptance of this paper as it is, but I would love if the authors could comment on my more "philoshopical" questions above, and even consider adding part of those reflections to their discussion.

**Special Issue:**

yes

---

### Official Review · AnonReviewer2 · 2021-03-09

**Confidence:** 5
**Preliminary Rating:** 3
**Recommendation:** Poster
**Final Rating:** 3

**Summary:**

This paper investigates a crucial question in the domain of unsupervised anomaly detection regarding the vulnerability of such methods to non-lesion based out-of-distribution (OOD) samples. Such samples which may indeed originate from a different domain compared to the training samples must be filtered during test phase because the model would not produce reliable results. The authors evaluate the ability of different metrics to distinguish in- and out-of distribution samples.

**Strengths:**

-Well written and clearly formatted
-Good and recent SOTA on unsupervised anomaly detection models based on generative models.
-Good and recent SOTA on OOD detection metrics.
-Comparison of different metrics (Dose WAIC etc) is well conducted.


**Weaknesses:**

- the paper does not address OOD samples in the training data. The main purpose is to detect OOD samples at the test phase.

-Section regarding pixel-wise anomaly detection (2.4) seems out of the main topic of the paper and should be removed

-Results of the performance analysis of the different metrics in Tables 1 and 2 demonstrate that the mean reconstruction error performs as well as more sophisticated metrics such as WAIC or the entropy based metric introduced by the authors.

-Some important illustrations, such as Figure 3, are located in the supplementary part but referred to in the main paper. In this form, i am not sure the paper format respects MIDL rules


**Deanonymize Review:**

no

**Detailed Comments:**

-In the SOTA, please also refer to unsupervised anomaly detection models that do not rely on GAN models or on the estimation of the reconstruction errors. Indeed, there is quite a large literature of methods based on the combination of deep neural networks for the extraction of  effective latent representations that serve as the inputs of anomaly detection models, such as Gaussian mixture models or Oc-SVM. [Seeböck, IEEE TMI 2018], [Alaverdyan, MEDIA 20],

-In sections 1 and 3 the authors mention that there exists no public brain MR datasets holding both healthy and lesional samples. The ADNI database, for instance, contains both types of data. Please reword this part of the text.

-The main formatting is not clear. Some results are presented in section 2.3. (First 4 lines), while the experiments are presented below. These lines aim, from what I guess, at arguing the need to introduce a novel metric, but they are confusing so I suggest to remove them.

-The main objective is to disentangle in- and out-of the distribution samples in the test set, why not in the training set ?

-Section 3.1 presents the performance analysis of the different metrics to disentangle different types of OOD testing data. It is not very clear how we should interpret performance metrics computed on the ‘lesion’ data of Table 1. As an example, the L1 measure for the Brats T2 HM lesion data allows achieving a very good AUC value of 0.92,  meaning that the model allowed discriminate Brats lesion images from normative CamCAN image. This however does not indicate whether these Brats images were considered positive because they contain a lesion or because they originate from a different distribution than the training dataset? My question is thus how we should handle such a result. The authors should clarify why this result is informative. Same comment apply to Table 2 which only consider ‘lesional’ slices.

-I would thus mainly focus on the performance achieved on the non lesional data of the BraTs and CamCAN dataset. Then my second question would be: Regarding AUC of the L1 reconstruction error, Table 1 indicates an AUC value of 0.51 for the CamCAN dataset and of 0.84 (0.85) for the Brats healthy data. I would deduce from this comparison that the healthy Brats datasets are non lesional OOD samples of the healthy CamCAN distribution. Based on this result, I would say that lesional data from the BRATS dataset can not reliably be evaluated with the GAN model (whatever the metric performance on the lesional BRATS data). Please comment .

-Results of the performance analysis of the different metrics in Tables 1 and 2 demonstrate that the mean reconstruction error performs as well as more sophisticated metrics such as WAIC or the entropy based metric introduced by the authors. This should be further discussed.

-Section 2.4 regarding the pixel-wise anomaly detection mainly presents a thresholding method of the reconstruction error maps. The link with the main objective of the paper, which is to disentangle lesional and non lesional OOD samples is not clear. Results on this section presented in Section 3.2 are mainly reported in the appendix section and do not add much because they are not compared with more standard thresholding methods and do not compete with SOTA performance, as stated by the author, because the GAN architecture is not optimal. Please clarify or remove this section.

-Regarding the description of the database in section 3, please add the number of training images of each class. For the BRATS Dataset, it is mentioned that images contain low and high grade gliomas, while later in the text, it is mentioned that you consider ‘healthy’ data. Please clarify.


**Final Rating Justification:**

I thank the reviewers for answering my questions and accounting for some of my comments. The authors address the very interesting topic of disentangling lesion-based OOD from their non-lesion based counterpart. They propose one novel OOD detection metric based on entropy, which however does not perform well. This limits the soundness of this paper.

**Justification Of The Preliminary Rating:**

The authors address the interesting question of the influence of OOD samples during the test phase of unsupervised anomaly detection models. The study is overall well conducted but some questions should be clarified to increase the soundness of this study.

**Paper Type:**

both

**Questions To Address In The Rebuttal:**

Please address all  detailed comments above.

**Special Issue:**

no

---

### Meta-Review · Area_Chair1 · 2021-02-22

**Recommendation:** Accept (Poster)

**Metareview:**

The paper aims to highlight the issue of unsupervised lesion detection being susceptible to false positive that stem from OOD variations that don't represent lesions. This doesn't come as any surprise, and, in fact, any one-class classification method would suffer from this limitation. This is an important problem, as has been noted by the reviewers.

Also noted by some reviewers is that fact that the paper doesn't really provide a rigorous methodological solution to the problem; a heuristic is provided in Section 2.3 which is only partly satisfactory.


**Paper Type:**

validation/application paper

---

### Decision · Program_Chairs · 2021-03-31

Accept